# Antiproliferative Aspect of Benzimidazole Derivatives’ Activity and Their Impact on NF-κB Expression

**DOI:** 10.3390/molecules24213902

**Published:** 2019-10-29

**Authors:** Katarzyna Błaszczak-Świątkiewicz

**Affiliations:** Department of Applied Pharmacy, Medical University of Lodz, Muszynskiego 1, 90–151 Lodz, Poland; katarzyna.blaszczak-swiatkiewicz@umed.lodz.pl

**Keywords:** cytotoxic activity, benzimidazole derivatives, bioreductive agents, breast cancer

## Abstract

Benzimidazoles belong to a new class of bioreductive agents with cytotoxic activity towards solid tumor cells, especially in their first stage of growth, which is characterized by low oxygen concentration. Bioreductive agents represent a class of prodrugs that target hypoxic tumor cells. Their bioactivity depends on the reactivity of their functional chemical groups. Their efficacy requires metabolic reduction and subsequent generation of toxic prodrugs. Chemoresistance of tumor cells is a major problem for successful antitumor therapy for many types of tumors, especially for breast cancer. The present study was performed to assess the effect of the antiproliferation activity of the tested benzimidazoles by way of NF-κB expression inhibition. The activity of the tested compounds on T47D and MCF7 cells was examined by WST, western blot, NF-κB transactivation assay, and apoptotic cell population analysis. Compound 3 was highly cytotoxically active against T47D cells, especially in hypoxic conditions. Its IC_50_ of 0.31 ± 0.06 nM, although weaker than tirapazamine, was significantly higher than the other tested compounds (2.4–3.0 fold). The increased bax protein expression upon exposure to the tested compounds indicated intercellular apoptotic pathway activity, with tumor cell death by way of apoptosis. Increased bax protein synthesis and apoptotic cell dominance upon treatment, especially with N-oxide derivatives (92% apoptotic cells among T47D cell populations during treatment with compound 3), were correlated with each other. Additionally, both increased bax protein and decreased NF-κB protein expression supported antiproliferative activity via NF-κB–DNA binding inhibition associated with the tested compounds. Compound 3 appeared to be the strongest inhibitor of NF-κB expression in hypoxic conditions (the potency against NF-κB expression was about 75% of that of tirapazamine). The present studies involving this class of heterocyclic small molecules proved their potential usefulness in anticancer therapy as compounds be able to limit tumor cell proliferation and reverse drug resistance by NF-κB repression.

## 1. Introduction

The lack of success in the search for breast cancer antitumor therapy has stimulated the scientific world to search for the pathway which will support the hormonal treatment of this pathology and enable an antiproliferation approach [1]. In many cases, traditional chemotherapy targets anti-apoptotic proteins, limiting recovery and even inducing drug resistance in the cancer [2]. One of the major agents which is highly expression during the invasion of many tumors, not just breast cancer, is NF-κB. NF-κB is active in inflammation and tumorigenesis, resulting from the stimulation of tissue proliferation and inhibition of programmed cell death in the apoptosis pathway [3,4]. NF-κB activity causes upregulation of anti-apoptotic protein genes and induction of cancer chemoresistance. Inflammation, which appears in cancer cells, is a complex physiological process which ensures the survival of the organism [5]. The transcriptional factor NF-κB has been found to be a major factor conjugated to IkB, an inhibitory protein involved in immune system action. When NF-κB is bound to IkB, its translocation to the nucleus is blocked. Degradation of this complex is performed via phosphorylation of IkB by IkB kinase, which follows inflammatory signal transduction and leads to NF-κB translocation to the nucleus. In the nucleus, NF-κB targets a DNA element and regulates the transcription of genes responsible for cell growth and apoptosis. Unfortunately, this protective mechanism of action is totally destroyed in cancer cells due to NF-κB overexpression, leading to the expression of anti-apoptotic genes and tumor cell invasion via uncontrolled proliferation [6,7,8,9,10,11].

Tests accomplished to date have provided evidence that bezimidazole derivatives can promote apoptosis and interrupt the cell cycle of A549 cells. Their toxicity profile, determined using healthy human erythrocytes and healthy human erythrocyte acetylcholinesterase enzyme (AChE), proved their potential biocompatibility and safe use with reference to AChE activity [12]. Therefore, following the studies previously performed with this class of compounds as bioreductive agents dedicated to antitumor therapy, especially for solid tumors with hypoxia, the present tests were conducted to determine their potential usefulness in an antiproliferation approach. To continue and expand on the current knowledge about the activity of new potential drugs among the benzimidazole derivatives, experiments with breast cancer cell lines were set up. The assays with T47D and MCF7 cell lines were planned and conducted so as to compare the biological activities of the tested substances in both cell lines. Mainly, NF-κB expression analysis upon the influence of the tested benzimidazole derivatives was studied and correlated with bax and bcl2 protein expression, and apoptotic cell population in the treated cells.

## 2. Results and Discussion

### 2.1. Tested Compounds

The tested substances represent a new potential class of bioreductive agents and belong to the derivatives of benzimidazole. The structures of the tested substances are presented in Figure 1. Tirapazamine (T) was used as a reference compound based on its highly selective cytotoxic effect on tumor cells, especially in hypoxic conditions.

Benzimidazole derivatives were prepared using the procedures described previously [13]. 

### 2.2. Cytotoxic Activity

The WST test performed on both tested cell lines provided evidence for the selective activity of the tested N-oxide benzimidazoles. Contrary to the N-oxide benzimidazoles’ effect, the benzimidazoles that were not N-oxide-bound showed better activity in normoxic conditions. The nitro derivatives presented higher activity than their analogues with a chloro-substituent, especially in hypoxic conditions. Both tested cell lines were sensitive to treatment with the tested benzimidazoles, although the T47D cell line seemed to be more so. The strongest cytotoxic activity in hypoxic conditions was observed for compound 3 (IC_50_ = 5.5 ± 1.8 nM and 0.31 ± 0.06 nM for MCF7 and T47D, respectively). Moreover, compound 3 was similarly potent in normoxia, especially in T47D cells (IC_50_ = 3.2 ± 0.01 nM), which may suggest an important role of the nitro group in its mechanism of action. Although its cytotoxicity to T47D cells was strong, the IC_50_ normoxia/hypoxia ratio of 34.7 for MCF7 cells indicated the strongest affinity to MCF7 breast cancer cells. Compound 1 showed similar properties, with the IC_50_ normoxia/hypoxia ratio of 16.1 (Table 1 and Figure 2).

### 2.3. Western Blots and Real-Time PCR Analysis for Bax and bcl2 Genes

In order to confirm the cytotoxic activity of the tested benzimidazoles based on the induction of apoptosis, the expressions of bax and bcl2 proteins were detected. The visualization of the selected proteins was carried out by western blot analysis. Both normoxic and hypoxic conditions showed bax protein expression in both lines for all tested compounds. Tirapazamine induced bax protein expression only in hypoxic conditions and did not induce it in normoxic conditions (Figure 3A). The bax protein synthesis was correlated with high bax gene expression, especially upon treatment with compounds 1 and 3 in hypoxic conditions (Figure 3B). The western blot analysis was supported by quantitative analysis of bcl2/bax protein expression by PCR. In light of their pro-apoptotic and anti-apoptotic activity, the ratio of bcl2/bax gene expression was estimated and the tendency towards reduction of bcl2 gene expression was observed. The bax gene expression increased in parallel. Compound 3 and 1 tended to be the strongest agents with pro-apoptotic function, especially in hypoxic conditions. This means that N-oxide and nitro group likely play a main role in the cytotoxic activity of the tested benzimidazoles (Figure 3B).

### 2.4. Expresion of NF-κB

The tested compounds were analyzed in terms of repression or activation of the NF-κB pathway. NF-κB, as a strong anti-apoptotic factor, can promote cell resistance to antitumor therapy. Therefore the NfKB activity in the test-compound-treated MCF7 and T47D cells was examined. Significant inhibition was observed for compound 3 and compound 4 in hypoxic and normoxic conditions, respectively. The NfKb transcriptional activity was inhibited by N-oxide derivatives of the tested benzimidazoles, especially in hypoxic conditions. The comparison of the luciferase signal of tirapazamine and the tested N-oxide derivatives’ influence on NfKb expression revealed the strongest NfKB inhibition by compounds 3 and 1. Compounds 3 and 1 inhibited NF-κB expression by approximately 75% and 70%, respectively, compared tirapazamine activity (results obtained for IC_50_ = 0.3 nM). Contrary to hypoxic conditions, the benzimidazoles showed stronger activity in normoxic conditions. Compound 4 showed the strongest NF-κB–DNA-binding repression, and the efficacy of compound 2 amounted to 62% compared to compound 4 (results expressed for IC_50_ = 0.3 nM) (Figure 4).

### 2.5. Evaluation of Apoptotic Cell Population and NF-κB Expression

Based on the cytotoxic activity of the tested compounds, the pathway of tumor cell death was estimated using the annexin/propidium iodide (PI) assay. Due to the problem of establishing the ratio of apoptotic to necrotic cells, and also in order to specify the selective activity for one of the tested cell lines, the indicated test was carried out. The obtained outcomes showed a dominant role of the early apoptotic cells, according to the rest of the cells in necrosis or late apoptosis for both tested cell lines. A higher apoptosis rate (between 92–93%) was detected for T47D cells, which could suggest a higher affinity of N-oxide derivatives to these tumor cells, especially in hypoxic conditions. The same N-oxide benzimidazoles (compound 3 and 1) induced a slight decrease in early apoptotic cells in the MCF7 cell line (84–87%). Benzimidazoles 4 and 2 displayed a similar activity in apoptosis indication in both cell lines, although the apoptotic cell percentage participation in the whole population was slightly decreased (75–93% for MCF7 and 53–65% for T47D in normoxia). When the apoptosis assay results were compared with the NF-κB expression results, the correlation between NF-κB transcription inhibition and cells apoptosis induction was concluded (Figure 5). N-oxide benzimidazoles proved their inhibition of NF-κB protein expression on the transcriptional level, especially in hypoxic conditions (Figure 5B). The compounds acted on NF-κB–DNA binding blocking in both cell lines, but their activity in MCF7 cells seemed to be stronger. A similar activity was observed for benzimidazoles. These compounds repress NF-κB protein synthesis, especially in normoxic conditions. They acted as strong inhibitors of NfkB expression in MCF7 cells, although their efficacy was much weaker than their analogues with an N-oxide bond. Out of the compounds tested in normoxic conditions, compound 4 showed the strongest affinity to NF-κB. Its activity was 1.5-fold stronger than the weakest compounds, 1 and 2, in normoxic conditions. Compound 3 was 2.5-fold more active than the weakest compounds, 2 and 4, in hypoxic conditions (Figure 5B).

## 3. Experimental Section

### 3.1. Materials and Methods

*Cell lines:* T47D and MCF7 (human breast cancer) cell lines were sourced from Sigma-Aldrich, Poznan, Poland. *Plasmids: pGL4.27* [luc2P/minP/Hygro] vector coding reporter gene of luciferase *luc2P* and *pGL4.32* [luc2P/NF-κB-RE/Hygro] vector coding an NF-κB response element (NF-κB-RE) were purchased from Promega. *Cell culture medium composition*: for T47D and MCF7 cell lines: DMEM; 10% FBS, 2 mM L-glutamine, 100 U/mL penicillin, 0.1 mg/mL streptomycin, 50 mg/mL hygromycin were purchased from Thermo Fisher Warsaw, Poland, Biowest Zgierz, Poland, and Sigma-Aldrich, respectively. *Chemicals:* tirapazamine was purchased from Sigma-Aldrich. *Transfection reagents*: Lipofectamine was sourced from Thermo Fisher Scientific. *Lysate reagents*: M-PER Mammalian Protein Extraction Reagent and Health Protease Inhibitor Cocktail were sourced from Thermo Fisher Scientific. *Antibodies for western blot analysis; primary antibodies:* anti-bax or anti-bcl2 came from Cell Signaling Technology and goat anti-rabbit HRP-secondary antibody from Santa Cruz Biotechnology. *Other reagents for western blot analysis*: 5% non-fat milk, TBS, and Tween 20 were purchased from Sigma-Aldrich and BioRad Warsaw, Poland, respectively.

### 3.2. Cell Culture

Both cell lines were cultured in DMEM growing medium, which was supplemented with 10% heat-inactivated (sterilized) fetal bovine serum (FBS), penicillin (10,000 U/mL), streptomycin (10,000 µg/mL), and amphotericin B (250 µg/mL).

Cell cultures were maintained in normoxic conditions (37 °C and 5% CO_2_). Tests were performed within 24 h in normoxic and hypoxic conditions. Poor oxygenation was created by reducing oxygen concentration below 1% O_2_. Morphological changes of the cultured cells were estimated using a phase-contrast microscope at 100-fold magnification (OptaTech, Warsaw, Poland).

The statistical data have expressed as mean ± SD and were analyzed by two-way ANOVA.

### 3.3. WST Cytotoxic Assay

Cell viability was analyzed by WST-1 test (Millipore, Sigma-Aldrich) according to the manufacturer’s instructions. Cells were seeded in 96-well plates in the density that provided 70% confluency before treatment and cultured in normoxic and hypoxic conditions, respectively. The exposure to test compounds lasted 24 h. Cells were treated with test compounds in the concentration range of 0.01 nM–200 µM. The stock solution was prepared with DMSO and diluted with growing medium to produce the final single sample concentration of 0.2% DMSO. After incubation, WST reagent was added to the cells and the absorbance was detected after 2 h at 440 nm using a microplate reader (Synergy H1, Bio-Tek Warsaw, Poland). The percentage of cell viability related to the control was calculated as [A] test/[A] control × 100. IC_50_ represents cell response determined using Graphpad Prism software—free version.

### 3.4. Western Blot Analysis of the Expression of Bax and bcl2 Proteins

In order to isolate the proteins of interest, the cells were lysed. Protein extraction from the lysates was performed at a low temperature (on ice) and the protein concentration was estimated using the BCA method. Next, the protein samples were boiled with 5× concentrated sample buffer, separated by SDS-PAGE, and then electrophoretically transferred to a PVDF membrane (BioRad). The membranes were blocked in 5% non-fat milk for 2 h at RT, washed with TBS/0.1% Tween 20, and incubated overnight at 4 °C with the primary antibodies: anti-bax or anti-bcl2 polyclonal antibodies. Subsequent incubation with goat anti-rabbit HRP-secondary antibody allowed immunodetection to be performed using an enhanced chemiluminescence kit with Chemidoc (BioRad) membranes.

#### Real-Time PCR Analysis of Bax and bcl2 Gene Expression

An RNA isolation kit (Thermo Fisher) was used to extract RNA from cells according to the manufacturer’s instructions. The RNA concentration was measured using an Eppendorf Biophotometer, and the purity of isolated RNA samples was controlled by the detected ratio of absorption at 260/280 nm. Briefly, the cDNA synthesis was performed based on the RNA matrix. cDNA was generated with a Maxima First Strand cDNA Synthesis Kit (Thermo Fisher) according to the manufacturer’s protocol. An optimized process of RNA transcription (500 ng, 20 °C for 15 min, 50 °C for 45 min, 85 °C for 5 min) allowed cDNA samples to be obtained, which were kept frozen at −20 °C.

Quantitative bax and bcl2 gene expression analysis was performed based on the reaction of 25 ng/5µL cDNA, 5 μL Fast qPCR Kit Master Mix (Sigma-Aldrich, Poznan, Poland), and 0.5 μL TaqMan Gene Expression Assay (20×). As a control, the TaqMan assay (Thermo Fisher) for B-cell CLL/lymphoma 2 with bcl2 (Hs00608023_m1) and bax-associated X protein (Hs00180269_m1) and β-actin (Hs01060665_g1) was performed. TaqMan PCR assays were performed on a real-time PCR system (Thermo Fisher, Warsaw, Poland) in 96 well PCR plates (Thermo Fisher). The following thermal cycling was performed: 25 s at 95 °C, 40 cycles each for 5 s at 95 °C, and 35 s at 60 °C. Expression values were calculated using Sequence Detection System Software. Fold induction values (RQ) were calculated according to the equation 2 − ΔΔCt, where ΔCt represents the differences in cycle threshold numbers between the target gene and β-actin, and ΔΔCt represents the relative change in these differences between the examined and control cells (calibrator).

### 3.5. Transfection of T47D and MCF7 Cells with NF-κB Gene and Reporter Vector LUC

The transcription levels of genes was analyzed to determine the tested compounds’ influence on NF-κB activity. The tested cell systems comprised transient and stably transfected cells with NF-κB linked to a *LUC* reporter gene construct. Stably transfected cells reached better protein expression; therefore, only stably transfected cells were used as to perform the transfection assay.

Cell test system: the initial cell density (0.5 × 10^3^ cells per well) was grown in 6 well plates in transfection medium without antibiotics. Only confluent cells were used for transfection with appropriate concentration of NF-κB DNA, using lipofectamine according to the manufacturers’ instructions. Next, the stable transfectans were cultured in the growth medium containing antibiotics (hygromycin 600 µg/mL and 800 µg/mL for T47D and MCF7 cells, respectively). The selection of stably transfected cells was carried out based on their resistance to hygromycin, which was introduced into the transfected cells together with the NF-κB DNA; only cells growing in the medium carried the selected gene.

### 3.6. Transactivation Assay

Cells stably transfected with *NF-κB* were cultured with a density of 12.5 × 10^3^ per well in a 96 well plate, and were transiently transfected with *LUC* reporter gene using lipofectamine based on the manufacturer’s instructions (the DNA amount was adjusted accordingly). After 24 h, cells were treated with test compounds for the next 24 h. To estimate the tested substances’ activity, the IC_50_ values were determined in the concentration range 0.1–1000 nM, and their activity was compared with tirapazamine tested in the corresponding concentration range (0.1–1000 nM). For luminescence analysis of luciferase activity, a VICTOR™ X Multilabel Plate Reader—Perkin Elmer, Warsaw, Poland was used.

### 3.7. Flow Cytometry Analysis of Apoptosis

The ratio of apoptotic to necrotic cells was estimated by APC-conjugated annexin V/PI test using flow cytometry analysis (FACS Canto II, Becton Dickinson, Warsaw, Poland). Cells were seeded at a density of 5 × 10^5^ in a T25 flask for 24 h and treated with tested compounds for a further 24 h. Cell were rinsed with PBS and detached using accutase. The detached cells were resuspended in buffer (100 µL/1 × 10^5^ cells) which contained APC-conjugated annexin V/PI (4 µL/1 × 10^5^ cells) and incubated for 20 min at RT, in darkness. The annexin-V- and PI-stained cells were evaluated for apoptosis, necrosis, and late apoptosis or necrosis populations, respectively.

## 4. Conclusions

Resistance to endocrine therapy for breast cancer is a common problem for many ER-positive patients [14,15,16]. One of the main factors involved in this pathology is the NF-κB transcription factor, which stimulates progression to estrogen-independent tumor growth. Additionally, NF-κB regulates cell survival via integration of the proinflammatory cellular signals that allow an aggressive and invasive form of inflammatory breast cancer (IBC) to develop [17,18,19,20]. This type of tumor is characterized by a correlation between NF-κB expression and ER activity [21,22,23]. Qiu et al. proved that NF-κB inhibitor BAY11-7082 blocks NF-κB transcription, and ER-mediated induction of cell proliferation upon its activity was reversed [21]. These data are useful in the development of therapy strategies for IBC, with ER-positive breast tumors usually resistant to endocrine therapy [24,25,26]. In the line with this evidence, the search for new potential inhibitors of NF-κB is necessary.

In the interest of defining the potential usefulness of benzimidazoles in re-resistance to antiproliferation action, NfkB expression upon exposure to tested benzimidazoles was evaluated. In order to get to know more about the mechanistic action of the tested compounds, since the NF-κB factor site is present in the promoters of the *bcl-2* and *bax* genes, the expression of the bcl-2 and bax proteins were estimated, and the correlation between NF-κB and anti-apoptotic and pro-apoptotic protein expression was analyzed. In order to compare the selectivity of the tested compounds, T47D and MCF7 cell lines were selected based on the compounds’ role as key mediators of chemoresistance.

The presented data delivered evidence for the benzimidazoles’ cytotoxic activity, with stronger efficacy in MCF7 cells. Cross-comparison of N-oxide benzimidazoles vs. benzimidazoles showed a selective affinity of N-oxide derivatives in hypoxic conditions. The strongest activity among the group of N-oxide benzimidazoles revealed nitro derivatives of N-oxide benzimidazoles. Benzimidazoles showed cytotoxic potential upon treatment in normoxic conditions. Nitro derivatives were more potent. To conclude, N-oxide binding and nitro-substituence may potentially affect benzimidazole selectivity and affinity to tumor cells, especially in hypoxic conditions.

The cytotoxic effect was examined via bax and bcl2 protein expression. The obtained outcomes proved pro-apoptotic bax protein synthesis in the cells treated with the tested substances. Additionally, neither N-oxide benzimidazoles nor benzimidazoles influenced bcl2 protein expression. To conclude, the tested compounds stimulated the pro-apoptosis pathway in both normoxic and hypoxic conditions.

Once pro-apoptotic protein expression was proven, the NF-κB expression needed to be evaluated as intercellular cross-talk of NF-κB proliferation pathway inhibition. NF-κB DNA-binding repression was observed in both tested cell lines, although MCF7 cells showed significantly lower expression of NF-κB. The strongest inhibition of NF-κB protein synthesis was seen with N-oxide benzimidazoles in both tested cell lines, especially in hypoxic conditions. To conclude, N-oxide benzimidazoles seem to be potential candidates for further study in the field of NF-κB inhibitor testing.

Additionally, in the light of the pro-apoptotic action of the tested N-oxide benzimidazoles, the apoptotic cell populations were estimated. According to the pro-apoptotic protein expression and NF-κB protein expression inhibition, the apoptotic cell populations dominated the populations of T47D and MCF7 cells under exposure to the tested compounds. These data were in compliance with expectations, especially for hypoxic conditions.

Benzimidazoles, with the major role of N-oxide binding in their action, may trigger breast cancer cells by the inhibition of NF-κB and induction of the apoptosis pathway in the re-resistance approach. The presented outcomes complement results obtained so far and confirmed the tested compounds’ selective cytotoxic activity in three tested tumor cell lines (T47D, MCF7, and A549), especially in hypoxic conditions. For breast and lung cancer, benzimidazoles showed pro-apoptotic activity, with *bax* gene transactivation leading to the induction of bax protein expression. Their bioreductive mechanism of action, with nitroreductase gene activation [26] together with NF-κB expression inhibition, potentially indicates their pro-drug engagement into antiproliferation pathway, which may help in the abolition of tumors’ chemotherapy resistance.

Despite the interesting results, further study should be performed in order to discover the basis for the molecular mechanisms of action of the tested benzimidazole derivatives.

## Figures and Tables

**Figure 1 molecules-24-03902-f001:**
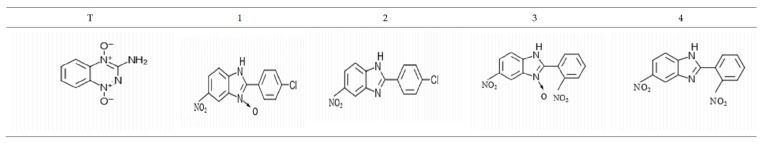
Structure of tested compounds 1–4 and tirapazamine (T).

**Figure 2 molecules-24-03902-f002:**
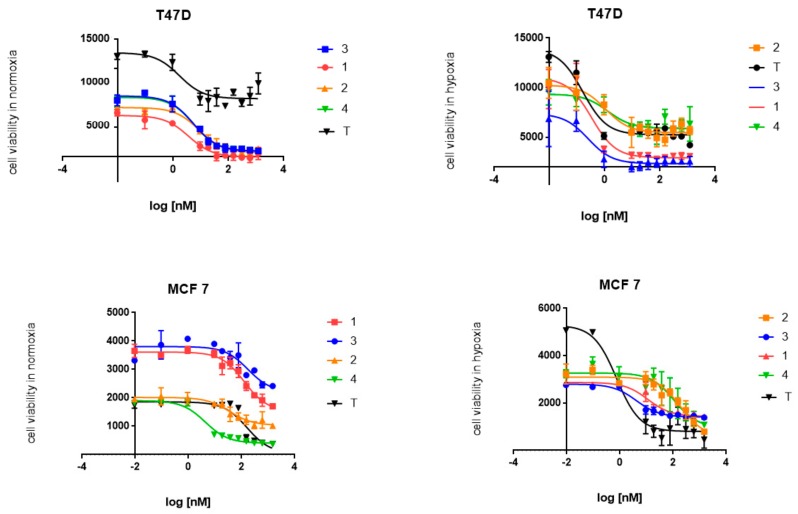
WST cytotoxicity assay; tested breast cancer lines (T47D and MCF7) were exposed to compounds 1–4 and tirapazamine (T); n = 3, data presented as mean ± SD.

**Figure 3 molecules-24-03902-f003:**
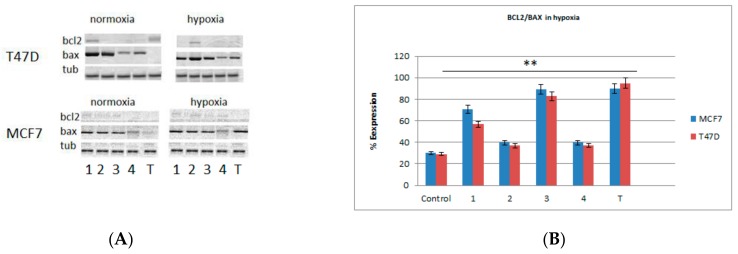
(**A**) Expression of bax and bcl2 protein biosynthesis under normoxic and hypoxic conditions in MCF7 and T47D cells; 1–4: samples with tested compounds; T: sample with tirapazamine as positive control. (**B**) Ratio of bcl2/bax gene expression in hypoxic conditions in MCF7 and T47D cells. 1–4: samples with tested compounds; T: sample with tirapazamine as positive control. Data are presented as mean ± SD, n = 3, *p* ≤ 0.01.

**Figure 4 molecules-24-03902-f004:**
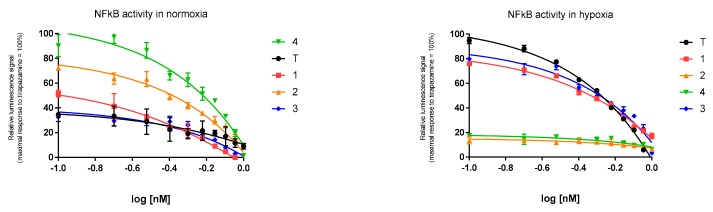
NF-κB transcriptional activity in T47D cells expressed by luciferase reporter gene assay after exposure to tested benzimidazole derivatives 1–4 and tirapazamine (T) for 24 h. Data have been normalized to tirapazamine and compound 4 maximal response for hypoxic and normoxic conditions, respectively; data represent significant difference at *p* ≤ 0.05 (n = 3, mean ± SD).

**Figure 5 molecules-24-03902-f005:**
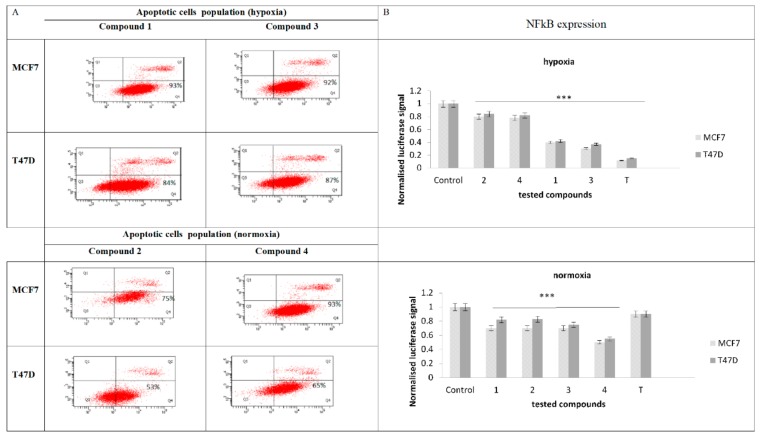
(**A**) Apoptotic cell population analysis for IC_50_ in T47D and MCF7 cell lines after 24 h incubation with tested compounds 1–4; (**B**) NF-κB expression analysis of IC_50_ by luciferase reporter gene assay after 24 h exposure to compounds 1–4, tirapazamine (T), and control luciferase gene reporter. Data come from three separate experiments performed in triplicate and are expressed as mean ± SD, **** p ≤* 0.001.

**Table 1 molecules-24-03902-t001:** Tested benzimidazole derivatives (1–4) and reference compound tirapazamine (T) and their cytotoxic effects on T47D and MCF7 cells in normoxic and hypoxic conditions. The data are presented as IC_50_ for n = 3, mean ± SD.

	**IC_50_ (nM) Normoxia [N]**				
	**1**	**2**	**3**	**4**	**T**
**T47D**	4 ± 0.05	4.3 ± 0.02	3.2 ± 0.01	3.5 ± 0.07	1.7 ± 0.06
**MCF7**	162.45 ± 2.2	13.5 ± 0.5	191.2 ± 3.3	8.5 ± 2.4	155.4 ± 1.2
	**IC_50_ (nM) hypoxia [H]**				
	**1**	**2**	**3**	**4**	**T**
**T47D**	0.33 ± 0.03	0.94 ± 0.05	0.31 ± 0.06	0.75 ± 0.02	0.12 ± 0.002
**MCF7**	10.1 ± 3.2	152.9 ± 2.7	5.5 ± 1.8	67.5 ± 1.9	2.45 ± 2.0
	**ratio [N/H]**				
	**1**	**2**	**3**	**4**	**T**
**T47D**	12.1	4.6	10.3	4.7	14.2
**MCF7**	16.1	0.09	34.7	0.12	63.4

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
