# Peer review of "Antiproliferative Aspect of Benzimidazole Derivatives’ Activity and Their Impact on NF-κB Expression"

_molecules, 2019, doi:10.3390/molecules24213902_

Round 1

Reviewer 1 Report

The manuscript of Błaszczak-Świątkiewicz describes the activity of a few (4) benzimidazoles in the presence of two breast cancer cell lines.

The study in itself is interesting but the manuscript must be improved before consideration for publication. In particular:

The English language must be revised e.g.

lines 8 and 9: with cytotoxic activity towards to solid tumor cells, especially in their first stage of growth proceeds with low oxygen concentration.

Line 11: was performed to assesses

Units are missing line 15 (IC50) Lines 69-70: ref 12 does not contain information on the synthesis of the benzimidazoles but send the reader to Błaszczak-Świątkiewicz K, Mirowski M, Kaplińska K, Kruszyński R, Trzęsowska-Kruszyńska A, Mikiciuk-Olasik E. New benzimidazole derivatives with potential cytotoxic activity— study of their stability by RP-HPLC. Acta Biochim Pol. 2012;59(2):279–88. Please use that citation. Quality of fig 1 should be improved and that is true for most figures; structures of the compounds in fig 1 cannot be read The authors should define “bioreductive agents” Line 89: fig 1 should be fig 2. In that fig, graphs could be shifted to a supplementary file. Line 101: fig 2 should be fig 3 Line 118: fig 3 should be fig 4 Page 7: fig 4 should be fig 5 Bibliography should be revised, many references are outdated The authors already studied the effects of those benzimidazoles on human lung adenocarcinoma A549 cells (ref 12). That must be mentioned and discussed in the introduction.

Author Response

Response to Reviewer 1 Comments

Point 1 : The English language must be revised e.g. :lines 8 and 9: with cytotoxic activity towards to solid tumor cells, especially in their first stage of growth proceeds with low oxygen concentration.Line 11: was performed to assesses.

Response 1: The English language was corrected in pointed out sentences and also the whole text was reviewed by English reviewer.

Point 2: Units are missing line 15 (IC50)

Response 2: The unit was added – [nM]

Point 3: Lines 69-70: ref 12 does not contain information on the synthesis of the benzimidazoles but send the reader to Błaszczak-Świątkiewicz K, Mirowski M, Kaplińska K, Kruszyński R, Trzęsowska-Kruszyńska A, Mikiciuk-Olasik E. New benzimidazole derivatives with potential cytotoxic activity— study of their stability by RP-HPLC. Acta Biochim Pol. 2012;59(2):279–88. Please use that citation.

Response 3: The suitable citation was adjusted – reference [13]

Point 4 Quality of fig 1 should be improved and that is true for most figures; structures of the compounds in fig 1 cannot be read

Response 4: The quality of figure 1 was improved. The figure 1 and 5 were shifted out to the supplementary file

Point 5 The authors should define “bioreductive agents”

Response 5: The bioreductive agent definition was implemented in abstract.

Point 6 Line 89: fig 1 should be fig 2. In that fig, graphs could be shifted to a supplementary file. Line 101: fig 2 should be fig 3 Line 118: fig 3 should be fig 4 Page 7: fig 4 should be fig 5

Response 6: The numbering of figures were adjusted

Point 7 Bibliography should be revised, many references are outdated

Response 7: The bibliography were checked, numbering was corrected and some new references were added

Point 8 The authors already studied the effects of those benzimidazoles on human lung adenocarcinoma A549 cells (ref 12). That must be mentioned and discussed in the introduction.

 Response 8: Some information about previously performed studies on A549 cells were mentioned in introduction, additionally summarize referring to tested A549 cell line and new breast cancer lines was described in conclusion.

Reviewer 2 Report

Whereas the work seems to be carefully done, some few points need attention before publication. Please find bellow some suggestions to improve the quality of the manuscript.

The authors should report a deeper discussion between cytotoxic activity and Western blot analysis.

In addition, a SAR study could also be reported in the current paper. This type of analysis can significantly enrich the paper.

In the Introduction section, it would be interesting to call attention for other classes of compounds used against cancer. For instance, 8-hydroxyquinoline derivatives containing a 1,2,3-triazole moiety or Mn2+ complexes of Benzothiazole Derivatives. In fact, a brief comparison among those classes could be interesting to justify the employ of Benzimidazoles toward to solid tumor cells. In this case, the following references could be cited:

EUROPEAN JOURNAL OF MEDICINAL CHEMISTRY   Vol. 84, 595-604 (2014)

BIOORGANIC CHEMISTRY   Vol. 84, 326-338 (2019).

LETTERS IN DRUG DESIGN & DISCOVERY   Vol. 14, 1415-1424 (2017).

CHEMISTRYSELECT   Vol. 4, 3118-3122 (2019).

JOURNAL OF MEDICINAL CHEMISTRY   Vol. 60, 2993-3001 (2017).

Author Response

Response to Reviewer 2 Comments

Point 1 : The author should report a deeper discussion between cytotoxic activity and Western blot analysis.

Response 1: The discussion between proteins and genes analysis was added , based on western blot and PCR deeper analysis

Point 2: In addition, a SAR study could also be reported in the current paper. This type of analysis can significantly enrich the paper.

Response 2: For sure this analysis would be helpful, however tested benzimidazoles structure was planned based on Tirapazamine known mechanism of action which targets DNA chain. So far performed tests proved benzimidazoles involvement into DNA function on the way of transcription and translation modification leading to certain genes and proteins expression.

Point 3.In the Introduction section, it would be interesting to call attention for other classes of compounds used against cancer. For instance, 8-hydroxyquinoline derivatives containing a 1,2,3-triazole moiety or Mn2+ complexes of Benzothiazole Derivatives. In fact, a brief comparison among those classes could be interesting to justify the employ of Benzimidazoles toward to solid tumor cells. In this case, the following references could be cited:

Response 3: The author absolutely agree with the reviewer that the discussion about two class of compounds would be curious, however it would be more relevant to compare both of them experimentally so as to emphasize their similarities and differences. The suitable reference has been cited.

Reviewer 3 Report

In this manuscript, Blaszczak-Swiatkiewicz examines the effects of benzimidazole derivatives on two cancer cell lines and NFkB expression. The aim of the author hold some value since this is an evolving area of research in the field but it suffers from several issues. There is a lack of details of methodology used for experiments, presentation of the results should be improved, and appropriate discussion should be provided. The manuscript is poorly written.

There are a few comments:

In figure 1, there are some cutoff of the chemical structure of Compound 1 and 2. Please provide the whole structures.

Extensive English language correction is required. Please remove double spaces! For instance, what does „apoptosis cells population analysis” mean?

Abstract – please provide the data, for instance, the IC50 values, not only …’was highly cytotoxic’. Add also units! Next – ‘an intercellular apoptotic’, can apoptosis be extracellular? Why tirapazamine is written in capital letter? ‘as a compounds’ - ??? These studies are only in vitro studies. The authors did not evaluated tumors. Please correct this issue. Abstract is poorly written, it should be structured into: introduction, methods, results and conclusions.

Introduction – why the author describes the effects of benzimidazole derivatives on AChE? How is it involved in the topic of the current paper?

The Author states that the synthesis of tested compound is presented in cyt. [12] but it is not true. The paper ‘Biological evaluation of the toxicity and the cell cycle interruption by some benzimidazole derivatives’ does not contain synthesis part.

The Author presents that these compounds exert anti-proliferative effects on MCF-7 cells at nanomolar concentration range. How is it possible, since in the previous publication of the Author the IC50 values were in μM? In the case of WST test was n only 3? How is it posssible? How the author determined the IC50 values, since on Figure 1 (should not it be figure 2?, please mark also which graph is for hypoxia and which is normoxia), lower, left graph the number of viable cells was not reduced by 50% (e.g. blue dots (comp. 3) the starting number is about 4000 cells, while the last point is ca. 2500. The same with tirapazamine on the lefr upper graph, starting with 15000 cells, the last measurement 10000). You cannot determine the IC50 value on the basis of such results. BTW, why do these graphs differ so much in the number of cells? In some of them there are 15000 and in the second one 5000 or 6000? How was the method validated?

In bax expression studies why do you not present results as mean and SD? What was n?

Did the authos conduct the experiments on primary cell line? It should be conducted!

Can you provide better resolution of figure 3 (btw it should be 4)? The axes description cannot be read. What does it mean that the data were normalized to tirapazamine and compound 4? Why these compounds? Please provide original data. What was the activity of NFkB at starting point (without tested compounds)? What about spontaneous control in these experiments?

In figure with apoptosis results, the FACS results are very difficult to understand. Those results are basely just the original output. The original data are needed to be analyzed and labeled clearly. These version of data should not be acceptable. Please clarify them. Control cytograms should also be presented. Figure 4 (5?)– why other significant relationships are not marked?

Materials and methods: what is this - L-glutamina, 100 U/ml penicylina, 0,1 mg/ml streptomycyna, 50mg/ml higromycyna?

The authors need to provide more information about the methodology used for the flow cytometry. The authors did not mention the staining regime in the methodology section itself. What about the apparatus?

Conclusion is not the exact conclusions. It resembles more a very basic discussion. It should be brief, and highlight the most important findings and possible implications.

‘The strongest activity among the group of N-oxide benzimidazoles revealed nitro-derivatives of N-oxide benzimidazoles’. – What does it mean?

Major revision is necessary before this manuscript can be considered for publication. The conclusion needs to be rewritten completely.

Please follow the guidelines provided by the journal for in text citations and the reference list.

I strongly suggest the authors to check their grammar and punctuation.

Author Response

Response to Reviewer 3 Comments

Point 1: In figure 1, there are some cut off of the chemical structure of Compound 1 and 2. Please provide the whole structures.

Response 1: The quality of figure 1 was improved

Point 2: Extensive English language correction is required. Please remove double spaces! For instance, what does „apoptosis cells population analysis” mean?

Response 2: The statement was corrected into ….’’apoptotic cells population analysis’’….

Point 3: Abstract – please provide the data, for instance, the IC50 values, not only …’was highly cytotoxic’. Add also units! Next – ‘an intercellular apoptotic’, can apoptosis be extracelular? Why tirapazamine is written in capital letter? ‘as a compounds’ - ??? These studies are only in vitro studies. The authors did not evaluated tumors. Please correct this issue. Abstract is poorly written, it should be structured into: introduction, methods, results and conclusions.

Response 3: The unit of IC50 was added; an intercellular refers to intercellular pathway, Tirapazamine – there is proper, not chemical name therefore there is in capital letter; tumor word was adjusted to tumor cell line; the abstract was written according to the editor requirements

Point 4: Introduction – why the author describes the effects of benzimidazole derivatives on AChE? How is it involved in the topic of the current paper?

Response 4: The short note referring AChE activity upon benimidazoles treatment was mentioned to emphasize their safety in vitro toxic profile

Point 5: The Author states that the synthesis of tested compound is presented in cyt. [12] but it is not true. The paper ‘Biological evaluation of the toxicity and the cell cycle interruption by some benzimidazole derivatives’ does not contain synthesis part.

Response 5: The proper reference was included

Point 6: The Author presents that these compounds exert anti-proliferative effects on MCF-7 cells at nanomolar concentration range. How is it possible, since in the previous publication of the Author the IC50 values were in μM? In the case of WST test was n only 3? How is it posssible? How the author determined the IC50 values, since on Figure 1 (should not it be figure 2?, please mark also which graph is for hypoxia and which is normoxia), lower, left graph the number of viable cells was not reduced by 50% (e.g. blue dots (comp. 3) the starting number is about 4000 cells, while the last point is ca. 2500. The same with tirapazamine on the lefr upper graph, starting with 15000 cells, the last measurement 10000). You cannot determine the IC50 value on the basis of such results. BTW, why do these graphs differ so much in the number of cells? In some of them there are 15000 and in the second one 5000 or 6000? How was the method validated?

Response 6: Previously published data refer to the A549 cell line, therefore the sensitivity of different tested cell lines upon tested benzimidazoles differ each other. The IC50 values were determined using GraphPad software –perfect tool for in vitro calculation . The cell lines differed between each other during culturing, therefore T47D cells were giving higher density (up to 15000) and MCF7 up to 6000. Two calibration curves were established separately for each cell line.

Poin 7: In bax expression studies why do you not present results as mean and SD? What was n?

Response 7: It was corrected

Point 8:Can you provide better resolution of figure 3 (btw it should be 4)? The axes description cannot be read. What does it mean that the data were normalized to tirapazamine and compound 4? Why these compounds? Please provide original data. What was the activity of NFkB at starting point (without tested compounds)? What about spontaneous control in these experiments?

Response 8: The results presented on Figure 4 represent NFkB expression on transcription level and there were calculated based on the strongest compound signal in tested group.

Point 9: In figure with apoptosis results, the FACS results are very difficult to understand. Those results are basely just the original output. The original data are needed to be analyzed and labelled clearly. These version of data should not be acceptable. Please clarify them. Control cytograms should also be presented. Figure 4 (5?)– why other significant relationships are not marked?

Response 9: The FACS analysis was performed according to standard procedure and represented data were included so as to emphasize the activity of tested compounds, control was performed as well (data not presented)

Point 10: Materials and methods: what is this - L-glutamina, 100 U/ml penicylina, 0,1 mg/ml streptomycyna, 50mg/ml higromycyna?

Response 10: These materials are required to maintain stably transfected cells

Round 2

Reviewer 1 Report

Quality of Fig 1 must be improved, formula cannot be read.

Reviewer 3 Report

In this manuscript, Blaszczak-Swiatkiewicz examines the effects of benzimidazole derivatives on cancer cell lines and NFkB expression. The aim of the author hold some value since this is an evolving area of research in the field but it suffers from lack of novelty and mechanistic insight. The author responded to some of my remarks, but unfortunately not to all.

There are still some major issues:

The manuscript has been poorly written. It should undergo a language correction (the whole manuscript).Double spaces have not been removed. The abstract has not been structure into: introduction, materials and methods, results and conclusions. The effects of benzimidazole derivatives on AChE cannot constitute a toxicity assessment of the tested compounds. The concentrations of thested compounds were not chosen properly. The author is not able to calculate IC50 values (even the program GraphPad Prism) when the data is not appropriate. The author made a mistake since she cannot count IC50 value taking into consideration the starting cell number (4000), and after co-incutation with compound (2500). There is not a half reduction! How was the method validated ? Can you give us a coefficient of variability? Was n = 3? If yes, it is not reliable. Did the authos conduct the experiments on primary cell line? It should be conducted! In addition, the author even did not try to explain why she did not conduct these experiments. Figure 5 is not present in the provide manuscript. Where are the control cells? Why the author do not want to add it in the figure? The author did not correct the reagents. Please change the language. We don’t have in English higromycyna etc.! The resolution of Figure 4 needs to be improved. The author did not respond to the following issues raised by me previously: The authors need to provide more information about the methodology used for the flow cytometry. The authors did not mention the staining regime in the methodology section itself. What about the apparatus? Conclusion is not the exact conclusions. It resembles more a very basic discussion. It should be brief, and highlight the most important findings and possible implications. ‘The strongest activity among the group of N-oxide benzimidazoles revealed nitro-derivatives of N-oxide benzimidazoles’. – What does it mean? Major revision is necessary before this manuscript can be considered for publication. The conclusion needs to be rewritten completely. Please follow the guidelines provided by the journal for in text citations and the reference list. I strongly suggest the authors to check their grammar and punctuation.